# Research on Influence of Tank Sloshing on Ship Motion Response under Different Wavelengths

Xinsheng Fan [ORCID], Zhenhong Hu and Xing Zheng *

College of Shipbuilding Engineering, Harbin Engineering University, Harbin 150001, China
* Correspondence: zhengxing@hrbeu.edu.cn; Tel.: +86-451-8256-8147

**Abstract:** The motion of the liquid-carrying ship under waves is simultaneously affected by the external wave moment and the sloshing moment of the internal tank, which makes the motion of the ship more complicated. In order to explore the influence of tank sloshing on the ship motion response, the motion of the ship model with tanks at different wavelengths were simulated based on the CFD software. This paper is based on the finite volume method (FVM) to solve the RANS (Reynolds averaged Navier-Stokes) equation for numerical simulation, and the VOF (volume of fluid) method was used to capture the free surface. Using the VOF method, it is necessary to ensure that the mesh at the free surface is sufficiently fine. Based on the above conditions, the pitching motion and rolling motion of the liquid carrier in the head sea condition and the transverse wave condition were simulated, respectively. The results showed that the sloshing of the tank has little influence on the pitching motion but has a greater influence on the rolling motion. When the liquid loading rates were 0, 0.8, 0.9, and 0.98, the pitch angle changes of the ship under different wavelengths were basically the same. However, when the liquid loading rates were 0, 0.4, and 0.8, the roll angle of the ship varied greatly under different wavelengths. By simulating the roll free decay motion of the liquid carrier (the loading rates were 0, 0.2, 0.4, 0.6, 0.8 and 0.98), it was found that the essence of the sloshing effect of the tank is to change the amplitude of the ship's rolling motion by changing the natural frequency of the ship's rolling motion. The closer the incident wave frequency is to the natural frequency of the roll of the liquid carrier, the greater the roll amplitude of the ship.

**Keywords:** tank sloshing; ship movement; regular waves; overlapping grid; natural frequency

## 1. Introduction

As the economy develops, the demand for liquefied natural gas (LNG) is increasing. However, the distribution of LNG in the world is not balanced, and many countries need to import a large amount of LNG every year. LNG ships are civilian ships specially designed to carry LNG. During the transportation of LNG ships, the hull will be subjected to external waves, which will cause pitch, roll, and other movements. The movement of the hull will drive the tank to move together, resulting in the occurrence of tank sloshing. The moment generated by the sloshing of the tank in turn changes the motion of the ship, making the motion of the ship uncertain. Therefore, it is particularly important to explore the influence mechanism of tank sloshing on ship motion.

The research on tank sloshing includes theoretical research, experimental research, and numerical research. Yang [1] carried out model test research on large-scale tanks, made statistical analysis on the distribution law of peak sloshing load, and compared several fitting distribution models. However, the cost of model testing is high, and the applicability is limited. The applicability of theoretical research is even more limited, and it is only suitable for some simple models. It is difficult to obtain analytical solutions for some strong nonlinear problems, including wave-breaking. In recent years, with the development of computer technology, computational fluid dynamics (CFD) has become more and more popular due to its universally applicable and low-cost characteristics. The

numerical solution obtained by the CFD method can approximately describe the fluid motion and the force of the object in the entire computational domain, making up for the limitations of the theoretical research method and the experimental research method. Wu [2] established a two-dimensional horizontal tank numerical model based on the VOF method to study the response of different external excitations to tank sloshing. Sun [3] combined the frequency domain solution of the ship motion and the tank sloshing hydrodynamic coefficient, established the ship frequency domain motion equation considering the tank sloshing, and explored the influence of the liquid carrying rate of the tank on the ship motion response. Zhuang [4] used OpenFOAM to simulate the coupling of FPSO ship motion and tank sloshing. In order to explore the influence of tank sloshing on the FPSO ship's rolling motion response, the impact pressure and generated moment of tank sloshing were studied. By calculating the time-domain equations of ship motion and tank sloshing and using the IRF method to obtain the ship's wave radiation force, the influence of the truncation error was discussed by Wen [5]. Wang [6] used the boundary element method to simulate the three-degree-of-freedom motion of a two-dimensional barge under wave and sloshing loads in the time domain. Xiao [7] used the impulse response function method to predict the hull motion, simulated the tank-sloshing problem based on the CFD theory, and discussed the effect of the wall shear force on the tank sloshing. Chen [8] combined the MPS method with the GPU parallel acceleration technology, wrote in the CUDA programming language, developed the MPSGPU-SJTU solver independently, and carried out a numerical simulation of the sloshing of a three-dimensional LNG tank. Based on the potential flow theory, Li [9] proposed a time-domain hybrid method for calculating the motion of a liquid carrier under waves and explored the influence of the nonlinearity of the external flow field on the motion performance of the liquid carrier.

Tank sloshing involves gas–liquid two-phase flow, and the capture of gas-liquid interface is a difficult problem. In this paper, the volume of fluid (VOF) method is used to capture the free liquid surface. The VOF method is a popular interface capture algorithm. It was first proposed by Debar (1974) [10] and further studied by Hirt & Nichols (198l) [11]. It can be used to simulate violent sloshing. Many scholars have developed many methods to solve the sloshing problem based on the VOF method [12–24]. Shen [16] adopted the concept of some element parameters, improved the traditional VOF method, and then used the improved VOF method to carry out numerical simulations on the prismatic tank model under different filling levels and different rolling excitation periods. Loots [21] numerically simulated the sloshing problem in an LNG tank using an improved VOF method. Based on the software OpenFOAM, Wang [24] simulated the motion process of two-dimensional liquid and ice tanks in waves and compared the calculation results of the water-carrying tank and the ice-carrying tank with ice of equal mass.

In this paper, regular Stokes waves are generated by wave-making at the velocity inlet. The pitching and rolling motions of the liquid carrier in regular waves with different wavelengths are respectively calculated. In the head sea condition, this paper explores the influence of the liquid loading rate and the wavelength of the incident wave on the pitching motion of the ship. In the transverse wave condition, the influence of the liquid loading rate on the natural frequency of the rolling motion of the ship is explored. In addition, the influence of the natural frequency of the ship's rolling motion and the frequency of the incident wave on the amplitude of the rolling motion is also analyzed.

## 2. Basic Principles and Methods

### 2.1. Governing Equation

The sloshing of the tank and the movement of the ship under waves involve two different media, water and air, and belong to the gas–liquid two-phase flow problem. In this paper, the gas-liquid energy exchange and compressibility are not considered, so the governing equations are the continuity equation and the momentum equation of the incompressible fluid:

$$\nabla \cdot \vec{u} = 0 \tag{1}$$

$$\frac{\partial(\rho\vec{u})}{\partial t} + \nabla(\rho\vec{u}\vec{u}) = -\nabla P + \nabla \cdot (\mu(\nabla\vec{u} + (\nabla\vec{u})^T)) + \rho\vec{g} + \nabla\tau_t + \vec{F}_s \tag{2}$$

where $\vec{u}$ is the velocity of the fluid; $\rho$ is the density of the fluid; $P$ is the fluid pressure; $\mu$ is the dynamic viscosity coefficient of the fluid; $\vec{F}_s$ is the surface tension term, and $\vec{g}$ is the acceleration of gravity. $\nabla\tau_t$ is the Reynolds stress term, which makes the RANS equation not closed, and the k-$\varepsilon$ turbulence model is introduced to close the RANS equation. This paper uses the realizable k-$\varepsilon$ turbulence model and uses two layer all y+ wall treatment. In addition, the y+ number is controlled around 150 by adjusting the mesh size of the prism layer.

*2.2. Moving Grid Technology*

In the simulation of tank sloshing, the moving grid technology is needed because the tank boundary will move. The dynamic change process of the mesh has many forms, and the dynamic mesh can be divided into a local redrawing model, a dynamic layer model, and a spring smooth model. The core idea of the spring fairing method is to apply Hooke's law to calculate the force value according to the displacement of the node at the boundary. After iterative calculation, the position where the node resultant force is zero can be obtained, which is the new grid node position. Under the action of external force, the entire spring system is adjusted through continuous iterative updates to obtain a new force balance. This model is suitable for the sloshing problem of the tank.

In any control volume $V$, the integral conservation equation of the generalized scalar $F$ is

$$\frac{d}{dt}\int_V \rho F dV + \int_{\partial V} \rho F(\vec{u} - \vec{u}_g) \cdot d\vec{A} = \int_{\partial V} \Gamma\nabla F \cdot d\vec{A} + \int_V S_F dV \tag{3}$$

where $\rho, \vec{u}_g, \vec{u}$ represent the fluid density, the moving speed, and velocity of the moving grid, respectively; $S_F$ is the source term; $\partial V$ is the boundary of the control volume $V$, and $\Gamma$ is the diffusion coefficient.

The time derivative term can be written in the first difference format as:

$$\frac{d}{dt}\int_V \rho F \, dV = \frac{(\rho FV)^{n+1} - (\rho FV)^n}{\Delta t} \tag{4}$$

In the formula, $n + 1$ and $n$ represent the $n + 1$th and nth time steps, respectively, and $V^{n+1}$ at the $n + 1$ time step can be written in the following form:

$$V^{n+1} = V^n + \frac{dv}{dt}\Delta t \tag{5}$$

where $\frac{dv}{dt}$ is the time derivative of the control volume, which can be written as:

$$\frac{dV}{dt} = \int \vec{u}_g \cdot d\vec{A} = \sum_j^{n_f} \vec{u}_{g\cdot j} \cdot \vec{A}_j \tag{6}$$

where $n_f$ is the surface mesh number of the control volume; $\vec{A}_j$ is the area vector of the surface $j$; and $\vec{u}_{g,j} \cdot \vec{A}_j$ can be calculated by the following formula:

$$\vec{u}_{g,j} \cdot \vec{A}_j = \frac{\delta V_j}{\Delta t} \tag{7}$$

where $\delta V_j$ is the space volume swept by the control surface $j$ in the time interval of $\Delta t$.

*2.3. Overlapping Grid Technology*

Compared with the dynamic mesh, the mesh quality of the overlapping mesh is better, and it does not require mesh deformation due to the movement of the hull, so it is more

suitable for simulating the large movement of the ship. Its basic idea is to divide the computational domain into a background area and an overlapping area, and exchange flow field information with the background grid through the overlapping interface. Information exchange is generally realized by interpolation between the receiver unit and the contributing unit. The interpolation generally adopts Lagrangian linear interpolation, which has the following form:

$$\phi_r = \Sigma \varepsilon_i \cdot \phi_i \tag{8}$$

where $\phi_i$ is the physical quantity of the adjacent contributing unit; $\phi_r$ is the physical quantity of the receiver unit; and for three-dimensional problems, $\varepsilon_i$ is the shape function corresponding to the tetrahedron composed of the center of the adjacent contributing unit as the vertex.

*2.4. Numerical Wave Making and Wave Elimination Methods*

The numerical pool used in this paper is a cuboid pool. Waves are generated by defining a wave velocity function at the inlet boundary. By controlling the movement of the water quality point on this boundary, the free liquid surface changes periodically, thereby generating waves.

The wave in this paper is a first-order Stokes wave, and its waveform can be expressed as:

$$\eta = A \, \cos(kx - \omega t) \tag{9}$$

In the formula, $A$, $k$, $\omega$ are the wave amplitude, wave number, and circular frequency of the wave, respectively.

In addition, in order to reduce the influence of boundary wave reflection on ship motion, it is necessary to eliminate waves at the boundary of the pool. The wave elimination method used in this paper is the forced wave elimination method, which is realized by adding a source term to the discrete N-S equation:

$$q\varphi = -\gamma\rho(\varphi - \varphi^*) \tag{10}$$

where $\gamma$, $\rho$, $\varphi$, $\varphi^*$ are the forcing coefficient, fluid density, the current solution, and the forced solution of the transport equation, respectively. The forcing coefficient of the forced wave cancellation is not a constant. It varies smoothly from the zero point at the inner boundary of the forcing region to the maximum value at the outer boundary of the forcing region. It has the following form:

$$\gamma = -\gamma_0 \, cos^2(\pi x^*/2) \tag{11}$$

Cao Hongjian [25] and Zha Jingjing [26,27] have verified through a large number of examples that the mesh size at the free surface has a huge impact on the reliability of numerical wave generation. When meshing, the mesh size at the free surface should be adjusted to ensure wave-making accuracy. In addition, it should be ensured that there are at least 20 and 80 layers of grids in one wave height and wavelength range, respectively, and at least three wave height ranges should be set up with encrypted areas, which can effectively suppress the attenuation of wave amplitude and ensure the accuracy of wave making.

## 3. Numerical Model Establishment

In this paper, based on the CFD software, the CFD method based on viscous flow was used to calculate the motion of the liquid carrier in regular waves. The time discrete format is selected as second-order discrete. The time step was adjusted according to the period of the incident wave and controlled to be about 1/400 of the period of the incident wave.

In the calculation, we controlled the appropriate time step by observing the Cell Courant Number cloud image and controlled the Courant Number between 1–5. After many attempts, we arrived at a method in which we set a small time step at the beginning, about $10 \times 10^{-6}$, and increased the time step appropriately after the residual curve was

reduced to a small value ($10 \times 10^{-3}$) and stabilized to improve the calculation speed. At this time, the time step was about $1/400$ of the period of the incident wave. In addition, we adjusted the number of iterations through the time step size. On the basis of ensuring convergence, the number of iterations per time step was controlled at around 15. The separation flow solver based on the pressure method was used, the second-order discrete was selected for both time and space, and the gravitational acceleration was set to a constant $9.8 \text{ m/s}^2$. The liquid in the tank was water at $20\ ^\circ\text{C}$.

### 3.1. Introduction to Geometric Models

The ship model used in this paper is the KCS ship model, and the tank is a GTT-type tank, which adopts the form of four tanks in series. From the stern to the bow, the tanks are numbered as tank 1–tank 4, and the fore and aft ends of the tank are indented inward. The hull parameters are shown in Table 1.

**Table 1.** Main parameters of ship model.

| Parameter | Real Ship | Ship Model |
|---|---|---|
| Scale ratio | 1 | 37.9 |
| Length between vertical lines LPP/m | 230 | 6.0686 |
| Breadth B/m | 32.2 | 0.8496 |
| Design draft D/m | 10.8 | 0.2850 |
| Displacement $\Delta$/t | 51958.719 | 0.9560 |
| Vertical position of center of gravity KG/m | 7.2768 | 0.1920 |
| Longitudinal position of center of gravity LCG/m | 111.616 | 2.9450 |
| Pitch inertia radius Kyy/Lpp | 0.25 | 0.25 |
| Roll inertia radius Kxx/B | 0.4 | 0.4 |

The tank parameters are shown in Table 2.

**Table 2.** Main parameters of tanks.

| | Length (m) | Width (m) | Height (m) | Corner (m) | Indent Angle |
|---|---|---|---|---|---|
| Tank 1 | 0.5 | 0.667 | 0.417 | 0.2/0.1 | 10° |
| Tank 2 | 1.0 | 0.667 | 0.417 | 0.2/0.1 | 0 |
| Tank 3 | 1.0 | 0.667 | 0.417 | 0.2/0.1 | 0 |
| Tank 4 | 0.5 | 0.667 | 0.417 | 0.2/0.1 | 10° |
| Tandem region | 0.03 | 0.3 | 0.05 | 0 | 0 |
| Distance from tank centroid to bow L/m | | | 3.125 | | |
| Distance from tank bottom to baseline Z/m | | | 0.293 | | |

The geometry of the tank and the ship model are shown in Figure 1.

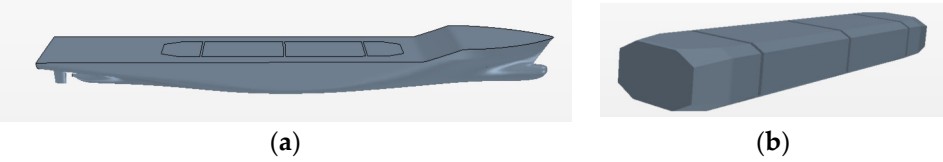

(**a**)          (**b**)

**Figure 1.** Geometric model of tank and ship. (**a**) Ship model; (**b**) tank model.

The pool is a cuboid, and there are differences in the size of the numerical pool under different working conditions.

In head sea conditions, the distance from the inlet to the bow is the greater of the length between length between perpendiculars (LPP) and the wavelength. The distance

from the exit to the stern is the greater of the length between 1.5 times the LPP and 1.5 times the wavelength. The distance between the side boundary of the pool and the shipboard is about equal to the LPP. The bottom of the pool is about 1.5 times the LPP from the free surface. The top of the pool is about one ship length from the free surface; see Figure 2a.

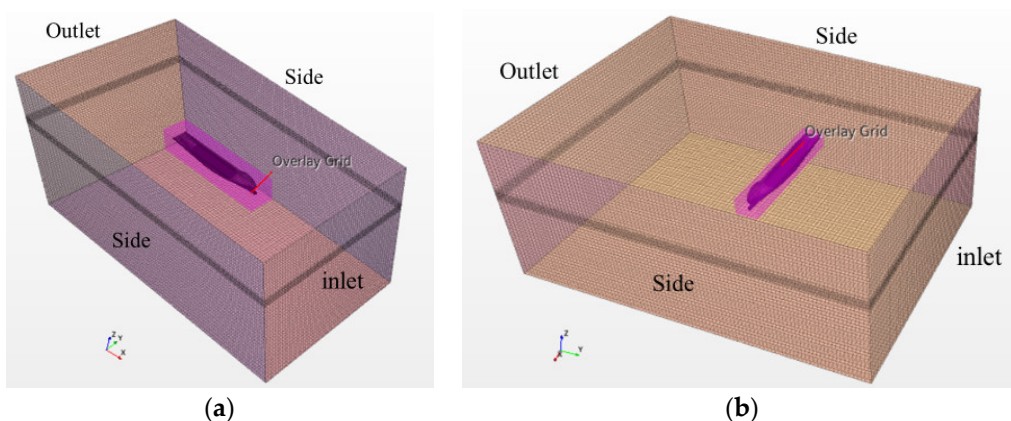

(a)  (b)

**Figure 2.** Numerical pool model. (**a**) Numerical pool in head sea condition; (**b**) numerical pool in transverse wave condition.

In transverse wave conditions, the distance from the inlet boundary of the pool to the shipboard shall be the greater of the length between the LPP and one time the wavelength. The distance from the exit to the shipboard is the greater of the length between 1.5 times the LPP and 1.5 times the wavelength. The distance between the side boundary of the pool and the ship's bow and stern is equal to the length of the ship. The bottom of the pool is 1.5 times the length of the ship from the free surface, and the top of the pool is about one ship length from the free surface; see Figure 2b.

### 3.2. Meshing Method

In order to ensure the accuracy of the calculation results, attention should be paid to the following:

(1) The number of prismatic layers on tank bulkhead and hull surface is not to be less than seven.
(2) Ensure that there are no less than 20 grids in a wave height range and no less than 80 grids in a wavelength range, and the grid slenderness ratio is not greater than 4.
(3) Due to the large curvature change near the hull, especially the bow and stern, the mesh here should be locally refined.
(4) The grid size of the overlapping area and the background area should not be too different to ensure more accurate information exchange between the two areas.
(5) The mesh should be locally refined near the free surface and in the upper half of the tank where slamming is likely to occur.

The meshing results are shown in Figures 3 and 4.

### 3.3. Empty Field Wave Verification

The wave environment of the calculated conditions was verified in a pool without a ship. The wave height value of the empty field wave and the theoretical value of the Stokes first-order wave were compared. In the head sea condition, the wave height measuring point was set at the bow. In the transverse sea condition, the wave height measuring point was set at the ship's side closer to the entrance. The comparison between the wave height of the numerical pool wave and theoretical value is shown in Figure 5.

It can be seen from the figure that the change of the numerical wave height is very close to the theoretical value. Therefore, the numerical pool at this grid size can meet the computational requirements and generate regular waves with high quality.

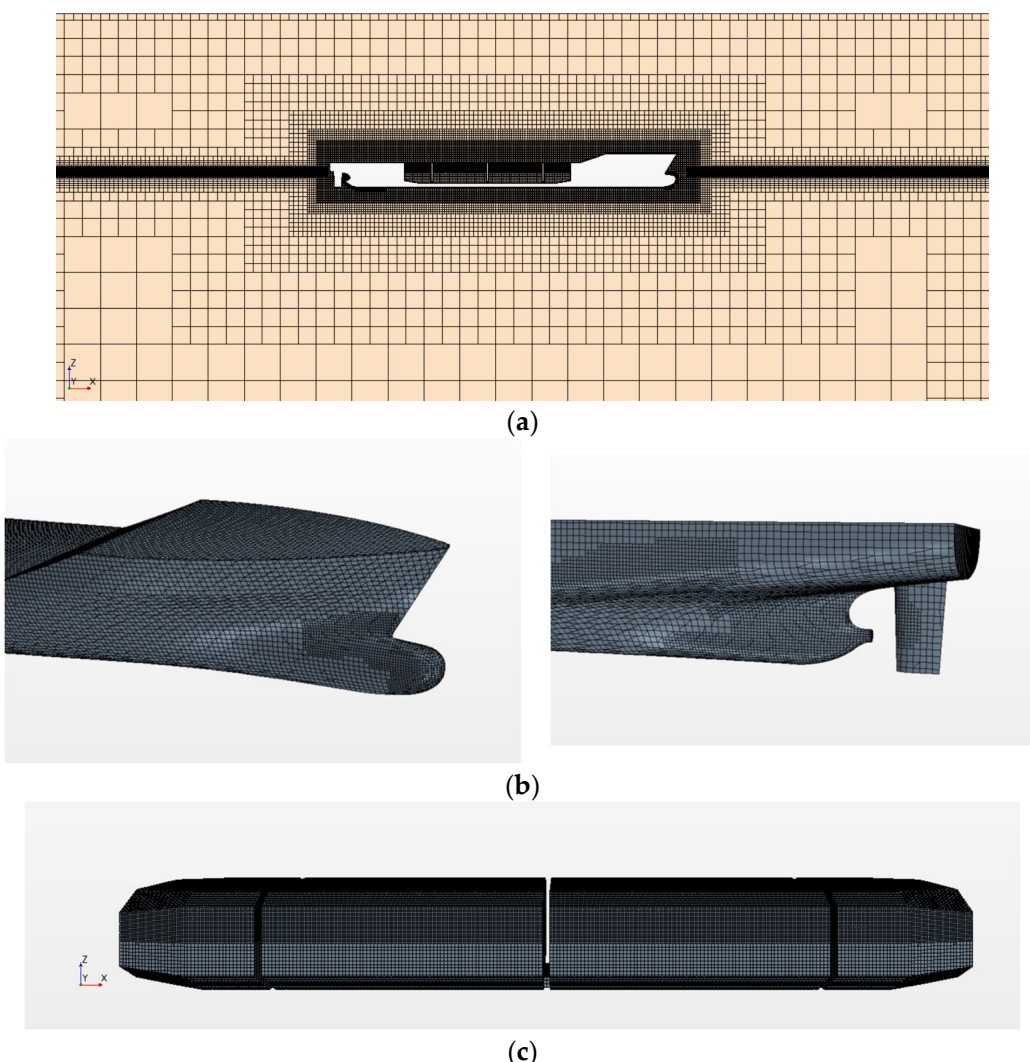

(**a**)

(**b**)

(**c**)

**Figure 3.** Meshing results in head sea condition. (**a**) Meshing of the mid-longitudinal section of the hull; (**b**) meshing of the hull fore and aft; (**c**) meshing of tanks.

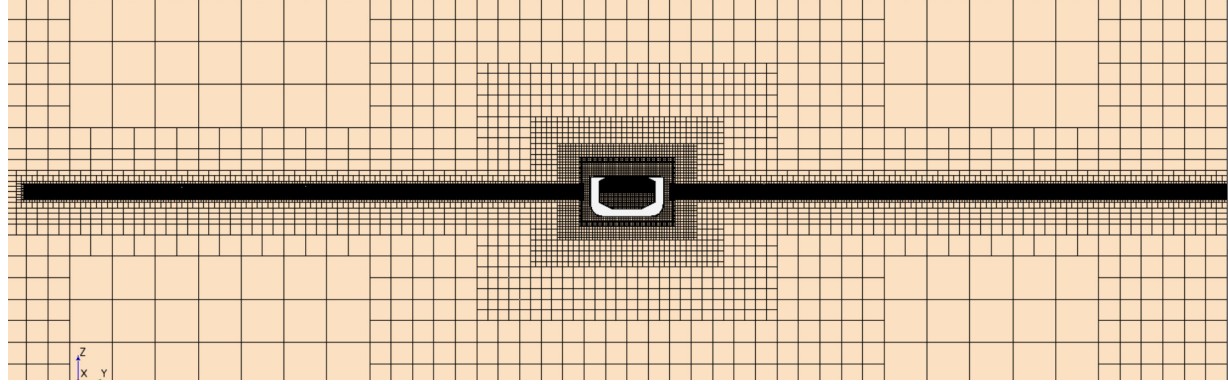

**Figure 4.** Meshing results in transverse wave condition.

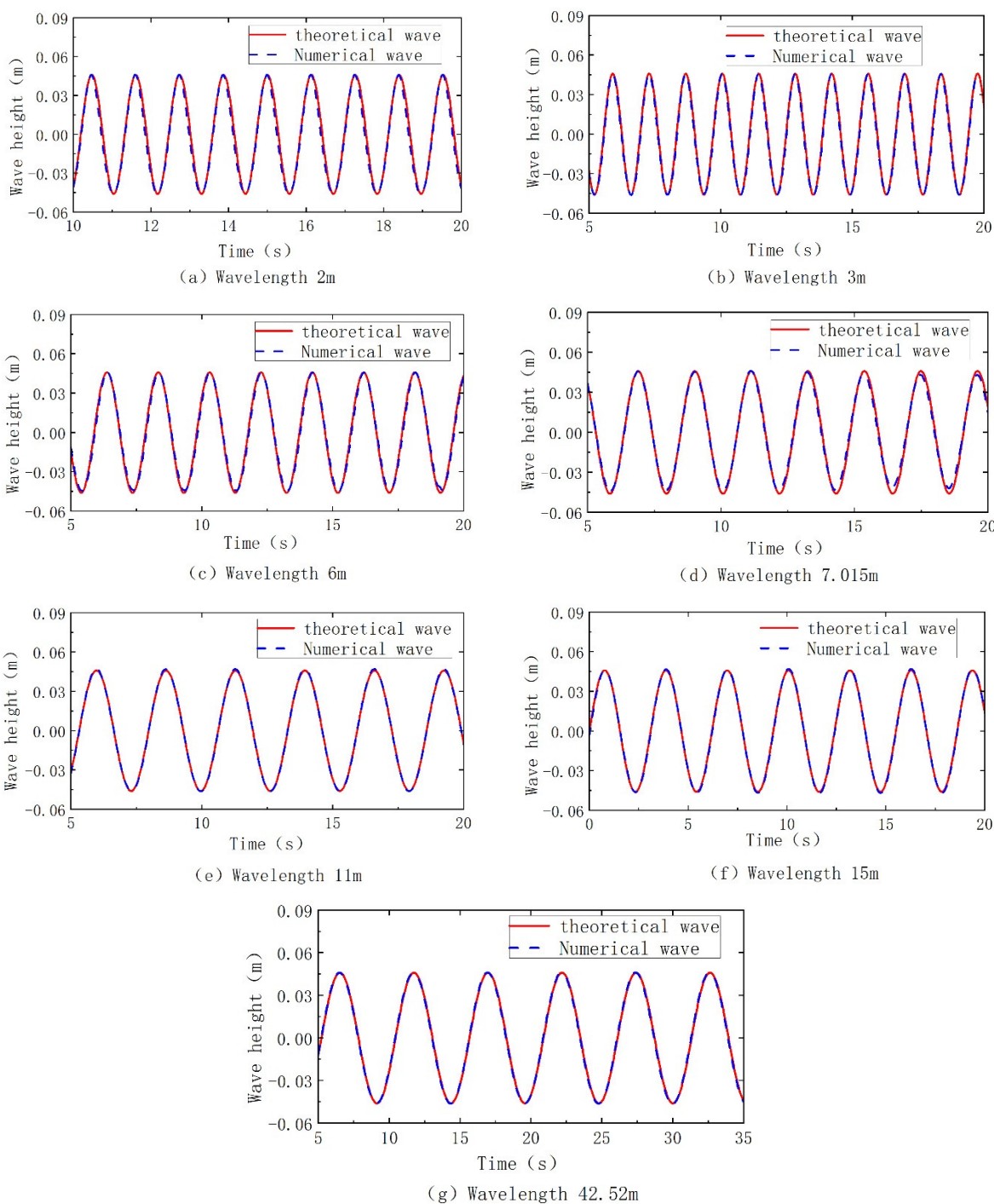

**Figure 5.** Comparison of numerical waves and theoretical waves.

### 3.4. Model Reliability Analysis

In order to verify the reliability of the model, the numerical calculation results are compared with the existing experimental results, or the results calculated by other methods. First, the numerical calculation of the pitch and heave motions of the ship under the condition of the head sea was carried out. Under this condition, the liquid loading rate of the tank was 0, and the wavelength of the incident wave was $\lambda = 7.015$ m ($\lambda/L = 1.15$). After nondimensionalization, stable periodic changes emerged; the results of ten cycles are analyzed here. The calculation results are compared with the experimental data of the 2015 Tokyo Workshop [28] as shown in Figure 6.

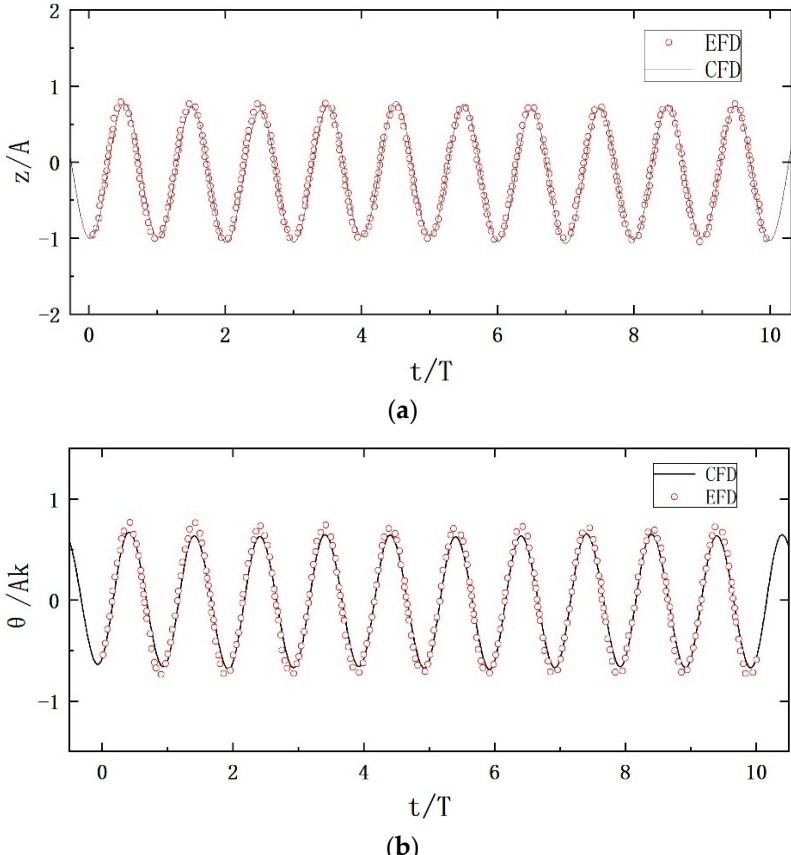

**Figure 6.** Comparison between numerical calculation results of ship motion and experimental values. (**a**) Ship heaving motion; (**b**) ship pitching motion.

Figure 6a,b are the results of dimensionless ship heave and pitch motions, respectively. It can be seen that the CFD calculation results based on the overlapping grid method are in good agreement with the experimental data, and the error is within the acceptable range. Generally speaking, the calculation results are reliable.

Subsequently, the rolling motion simulation of the liquid carrier was carried out, referring to the calculation results of Wen [4]. The model used is the FPSO experimental model made by the MARINE laboratory. In this paper, the same numerical model and meshing method were used to calculate the rolling motion of an FPSO with a liquid loading rate of 0.56 under a transverse regular wave with an incident wave frequency of 1.14 rad/s. The calculation results are compared in Figure 7.

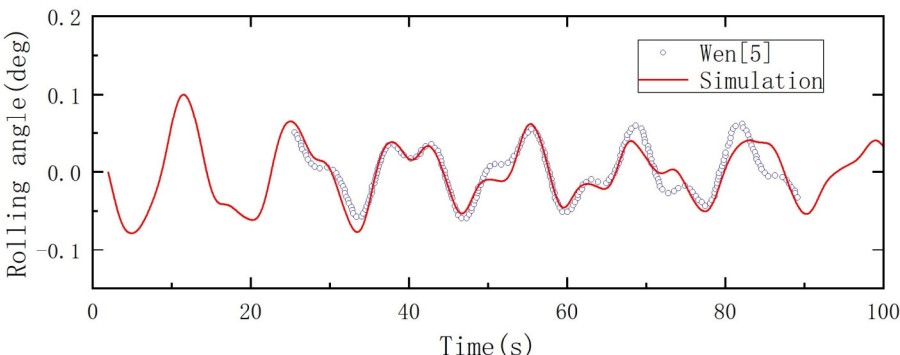

**Figure 7.** Verification of rolling calculation results.

It can be seen from Figure 7 that the CFD calculation results based on overlapping grids are in good agreement with the results in the literature as a whole. The change trend and cycle of the roll angle are basically the same, and the amplitude of the roll angle at different times is slightly different but within the acceptable range. In general, the numerical model and meshing method in this paper can well calculate the motion of the liquid carrier under regular waves.

## 4. Simulation of Pitching Motion of Liquid Carrier in Head Sea Condition

When the ship sails against the waves, it mainly has two degrees of freedom, heave and pitch. In order to eliminate the influence of the change of the ship's draft balance position due to different liquid loading rates, we only opened one degree of freedom for pitch, and always kept the ship at the design draft to explore the influence of tank sloshing on the ship's pitching motion.

### 4.1. Calculation Case Settings

The pitch motion of the ship with different liquid loading rates under different incident wave wavelengths was calculated when the Froude number $F_r = 0.26$. The calculation conditions are as follows:

### 4.2. Influence of Liquid Loading Rate on Pitching Motion of Ship

Cases 1–4 in Table 3 explore the effect of the liquid loading rate on the pitching motion of the ship under the same incident wave wavelength ($\lambda/L = 1.15$). Figure 8 compares the time history of the ship's pitching motion when the liquid loading rate is 0, 0.8, 0.9, and 0.98.

**Table 3.** Calculation condition settings in head sea condition.

| Case | Wavelength $\lambda$/m | Wave Height H/m | Degree of Freedom of Hull Movement | Loading Rate |
|------|------------------------|-----------------|-------------------------------------|--------------|
| 1 | 7.015 | 0.092 | pitch | 0 |
| 2 | 7.015 | 0.092 | pitch | 0.8 |
| 3 | 7.015 | 0.092 | pitch | 0.9 |
| 4 | 7.015 | 0.092 | pitch | 0.98 |
| 5 | 2.0 | 0.092 | pitch | 0.8 |
| 6 | 42.52 | 0.092 | pitch | 0.8 |

It can be seen from Figure 8 that under the condition of the same incident wave wavelength, the changes of the pitch angle of the ship with different liquid loading rates are basically the same. By observing the calculation results of a certain period, it can be seen that with the increase of the liquid loading rate, the pitch phase of the hull slightly shifts to the right. This is mainly due to the superposition of the moment generated by the sloshing of the tank with the external wave moment, which changes the phase of the original moment. However, due to the small sloshing moment of the tank, the contribution to the phase offset is limited, so that the phase of the pitch motion of the ship is only slightly offset. Figure 9 shows the liquid level change in the tank during pitching motion of the ship during one cycle when the liquid loading rate is 0.8. Figure 10 shows the pressure change in the tank.

In general, since the sloshing moment of the tank is much smaller than the external wave moment, the change of the liquid loading rate has little effect on the pitching motion of the hull. Therefore, it can be said that the effect of tank sloshing on the pitching motion of the hull is not significant.

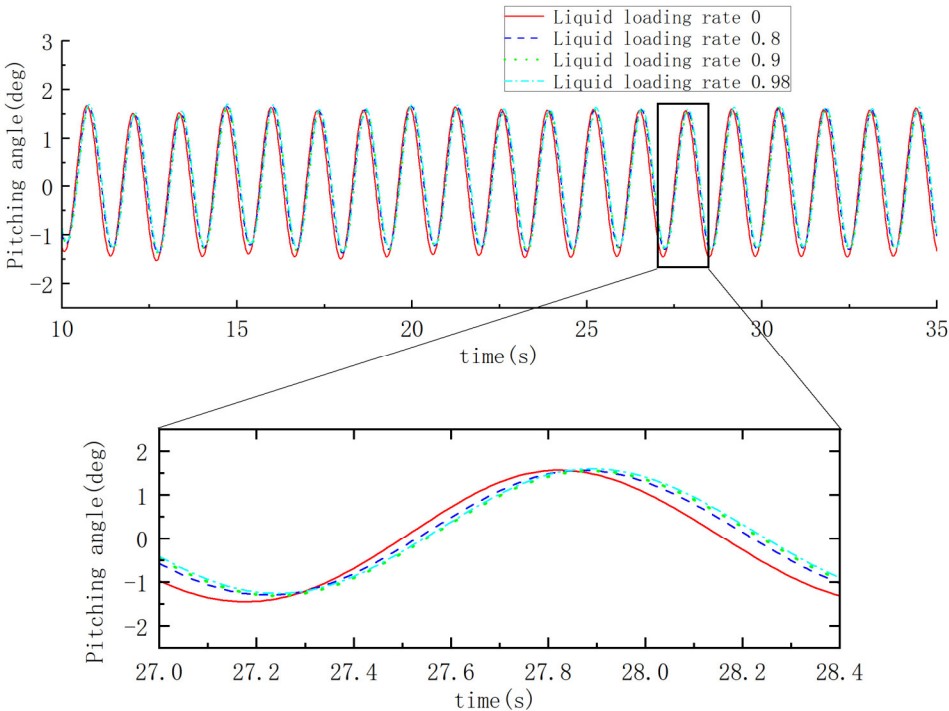

**Figure 8.** Time history curve of ship pitching motion under different liquid loading rates.

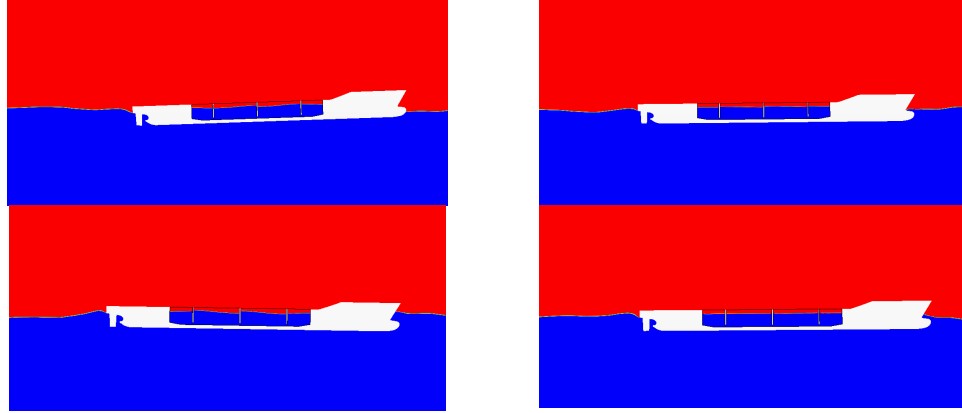

**Figure 9.** Change of liquid level in one cycle of ship motion with a liquid loading rate of 0.8.

*4.3. Influence of Wavelength of Incident Wave on Pitching Motion of the Liquid Carrier*

Cases 2, 5, and 6 explore the influence of the incident wave wavelength on the pitching motion of the ship when the liquid loading rate is 0.8. The pitch motion of the hull under different incident wave wavelengths is compared in Figure 11.

It can be seen from Figure 11 that when the wavelength of the incident wave is close to the length of the ship, the hull motion is the most violent (large amplitude and short period). In order to avoid sailing in waves with wavelengths close to the length of the ship, the frequency of the wave corresponding to the maximum energy value in the wave spectrum of the sea area can be obtained through the statistical value of waves in the sea area, and the wavelength $\lambda_0$ can be obtained through the dispersion relationship. When designing the ship, the captain should be set as far as possible from $\lambda_0$.

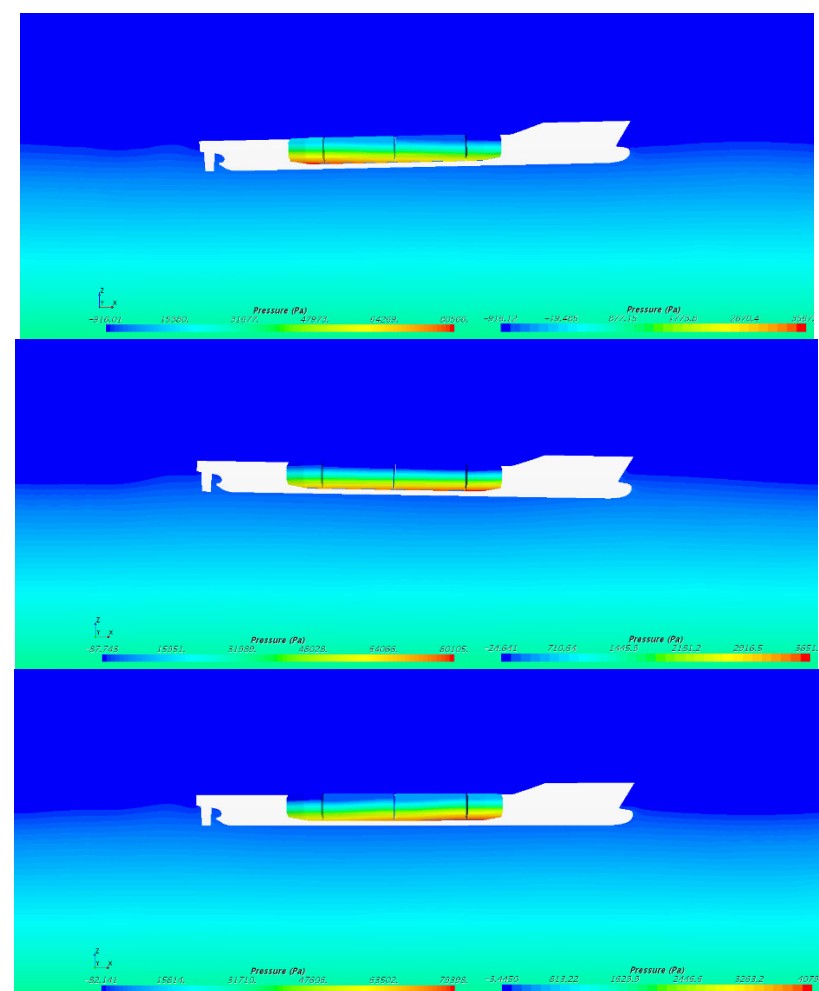

**Figure 10.** Variation of the pressure in the tank of the ship with a liquid loading rate of 0.8.

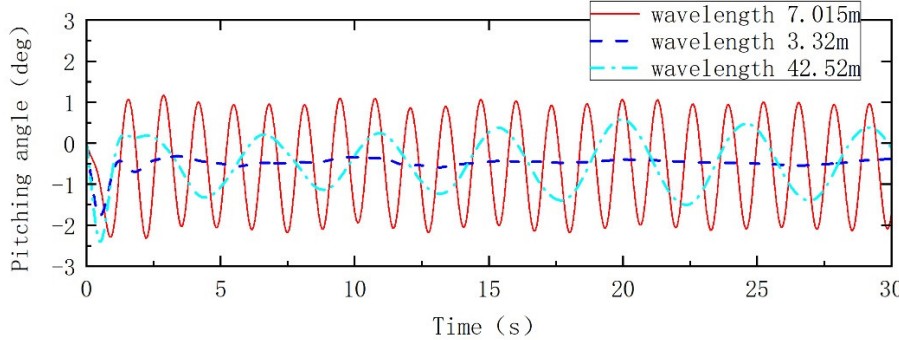

**Figure 11.** Pitch time history curves of ships with a liquid loading rate of 0.8 at different wavelengths.

## 5. Simulation of Rolling Motion of Liquid Carrier in Transverse Waves Condition

The ship mainly rolls and heaves in transverse waves condition. Also, to eliminate the effect of changes in the ship's draft due to changes in liquid loading rate, the ship should be set to roll only to explore the motion law of ships with different liquid loading rates under different incident wave wavelengths at zero speed.

### 5.1. Calculation Case Settings

In this part, the rolling motion of ships with different liquid loading rates under the same incident wave wavelength and the rolling motion of ships with a certain liquid

loading rate under different incident wave wavelengths were respectively calculated. The calculation conditions are set as Table 4:

**Table 4.** Calculation condition settings in transverse waves condition.

| Case | Loading Rate | Wavelength λ/m | Wave Height H/m | Degree of Freedom of Hull Movement |
|------|------|------|------|------|
| 1 | 0.8 | 3 | 0.092 | roll |
| 2 | 0.8 | 6 | 0.092 | roll |
| 3 | 0.8 | 11 | 0.092 | roll |
| 4 | 0.8 | 15 | 0.092 | roll |
| 5 | 0.4 | 3 | 0.092 | roll |
| 6 | 0.4 | 6 | 0.092 | roll |
| 7 | 0.4 | 11 | 0.092 | roll |
| 8 | 0.4 | 15 | 0.092 | roll |
| 9 | 0 | 3 | 0.092 | roll |
| 10 | 0 | 6 | 0.092 | roll |
| 11 | 0 | 11 | 0.092 | roll |
| 12 | 0 | 15 | 0.092 | roll |

*5.2. Moment Analysis of Liquid-Carrying Ship during Rolling Motion*

When the liquid carrier rolls, it is simultaneously affected by the external wave moment and the sloshing moment of the internal tank. The moment experienced by the ship under different working conditions is similar: the sloshing moment of the tank is always in the opposite direction to the external wave moment, and the periods of the two are consistent and equal to the rolling period of the ship. Figure 12 shows the moments experienced by the ship under some conditions.

It can be seen from Figure 12 that when the liquid loading rate is non-zero, the sloshing moment of the tank is always opposite in phase to the external wave moment, and the amplitude of the tank sloshing moment is smaller than that of the external wave moment. The superimposed resultant moment is also smaller than the external wave moment. When the liquid loading rate is 0, the sloshing moment of the tank is close to 0, and the external wave moment and the resultant moment curve basically coincide.

During the ship's rolling process, the liquid flow direction in the tank is always the same as the ship's rolling direction, the fluid in the tank is still dominated by static pressure, and the generated moment is consistent with the ship's rolling direction. Due to the large roll angle of the ship, the external moment mainly provides the restoring moment of the ship's roll, and its direction is opposite to the direction of the ship's rolling motion, so the sloshing moment of the tank is always opposite to the direction of the external wave moment. Figures 13 and 14 respectively show the changes of liquid level in the tank and the pressure in the tank when the ship rolls under some conditions.

*5.3. Influence of Tank Sloshing on Natural Frequency of Ship Roll*

In order to explore the effect of tank sloshing on the natural frequency of ship roll, the free decay motion of ships with different liquid loading rates in open water was simulated. By giving the ship an initial rolling angular velocity and recording the change of the rolling angle of the ship in the open water, the natural period of the rolling was obtained. The free decay motions of ships with liquid loading rates of 0, 0.2, 0.4, 0.6, 0.8, and 0.98 were calculated, respectively. Figure 15 shows the free roll decay curves of the ships with different liquid loading rates.

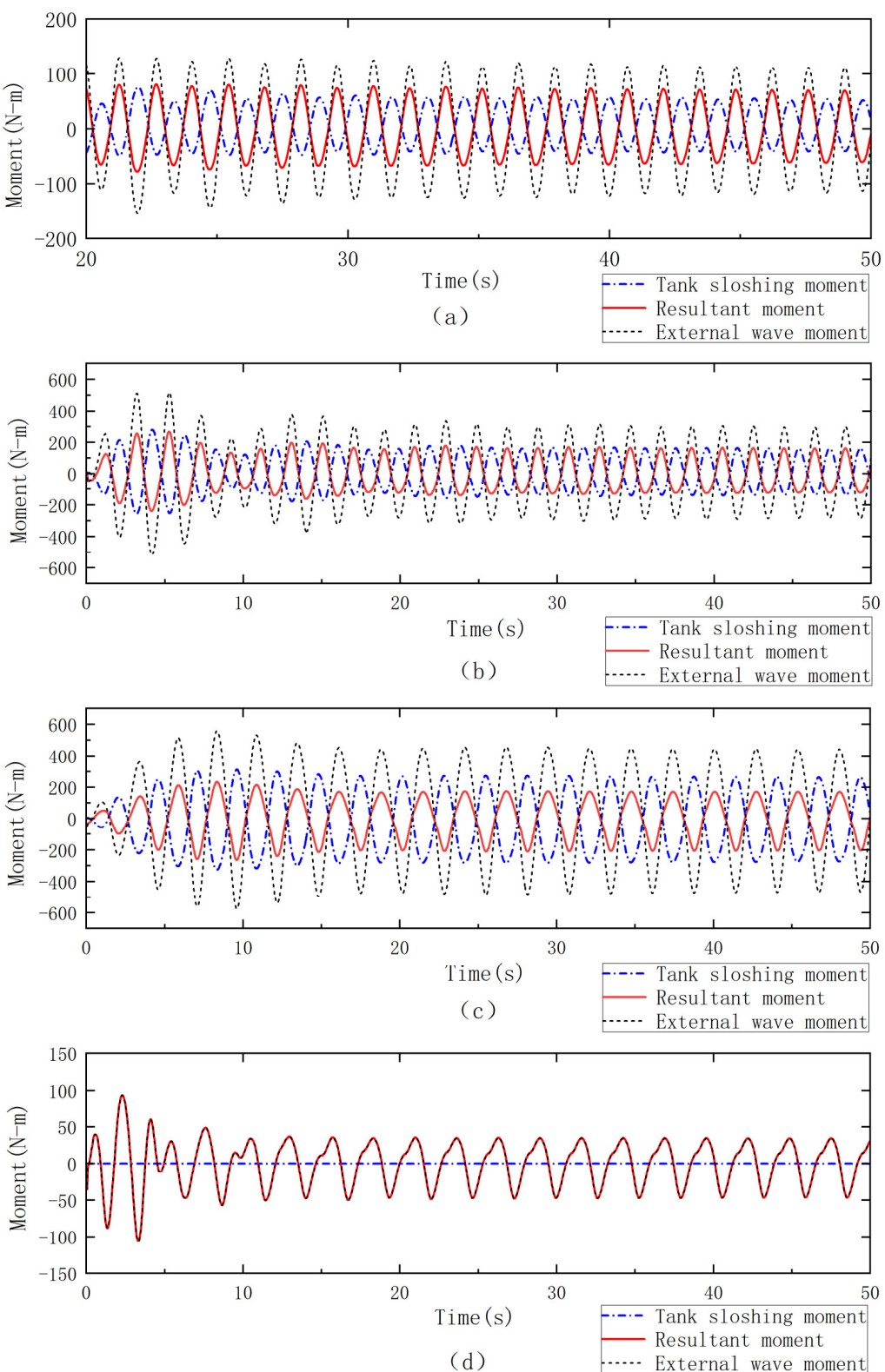

**Figure 12.** Moment of a liquid-carrying ship when rolling. (**a**) Liquid loading rate 0.8 and wavelength 3 m; (**b**) liquid loading rate 0.8 and wavelength 6 m; (**c**) liquid loading rate 0.8 and wavelength 11 m; (**d**) liquid loading rate 0 and wavelength 11 m.

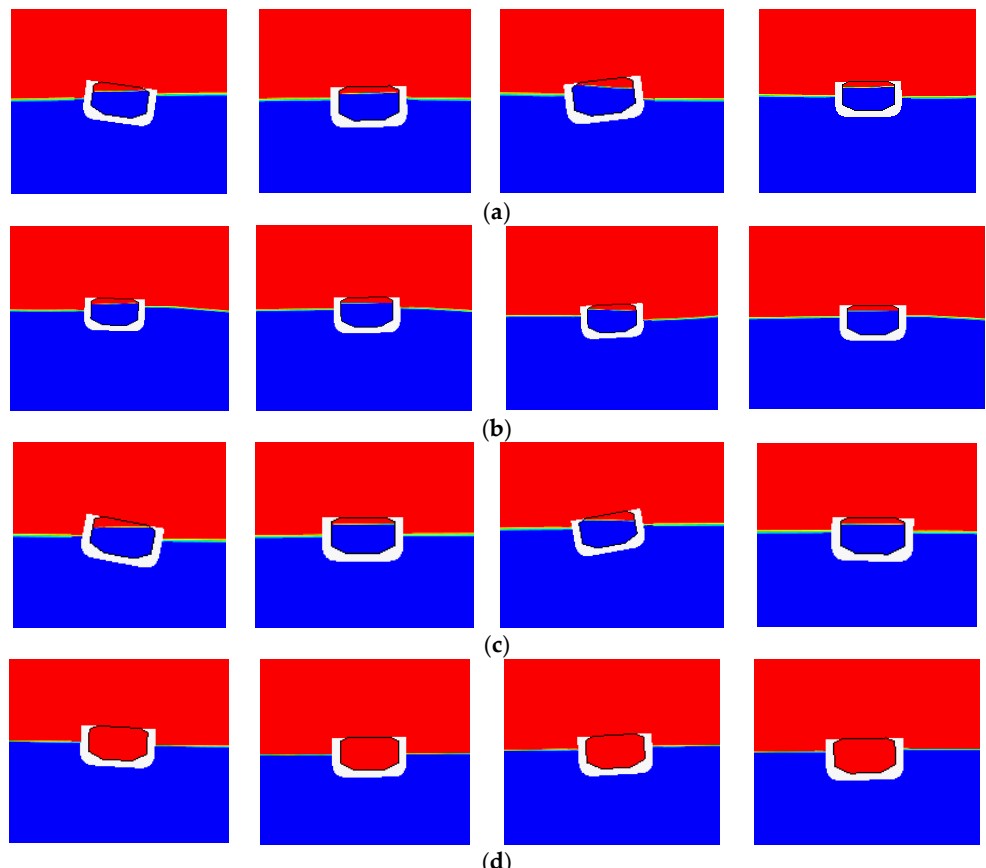

**Figure 13.** Changes of liquid level in tank during one period of ship rolling motion. (**a**) Liquid loading rate 0.8 and wavelength 6 m; (**b**) liquid loading rate 0.8 and wavelength 3 m; (**c**) liquid loading rate 0.8 and wavelength 15 m; (**d**) liquid loading rate 0 and wavelength 11 m.

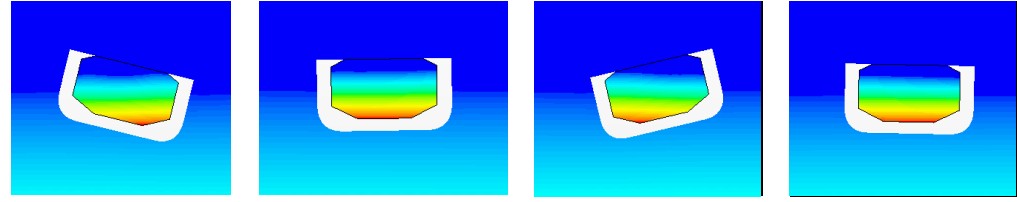

**Figure 14.** Variation of cabin pressure during one period of ship rolling motion.

Figure 15 compares the rolling free decay curves of adjacent loading rates in the design conditions. Next, according to Figure 15, we calculated the natural period of the ship's roll with different liquid loading rates when the roll angle is not less than 2° and obtained the natural frequency of the ship's roll, as shown in Table 5.

**Table 5.** Natural frequency of ship rolling motion with different liquid loading rate.

| Liquid Loading Rate | Ship's Initial Angular Velocity (rad/s) | Roll Natural Frequency (rad/s) |
|---|---|---|
| 0 | 1.2 | 3.634 |
| 0.2 | 1.2 | 2.729 |
| 0.4 | 1.2 | 2.705 |
| 0.6 | 1.2 | 2.659 |
| 0.8 | 1.2 | 2.417 |
| 0.98 | 1.2 | 2.927 |

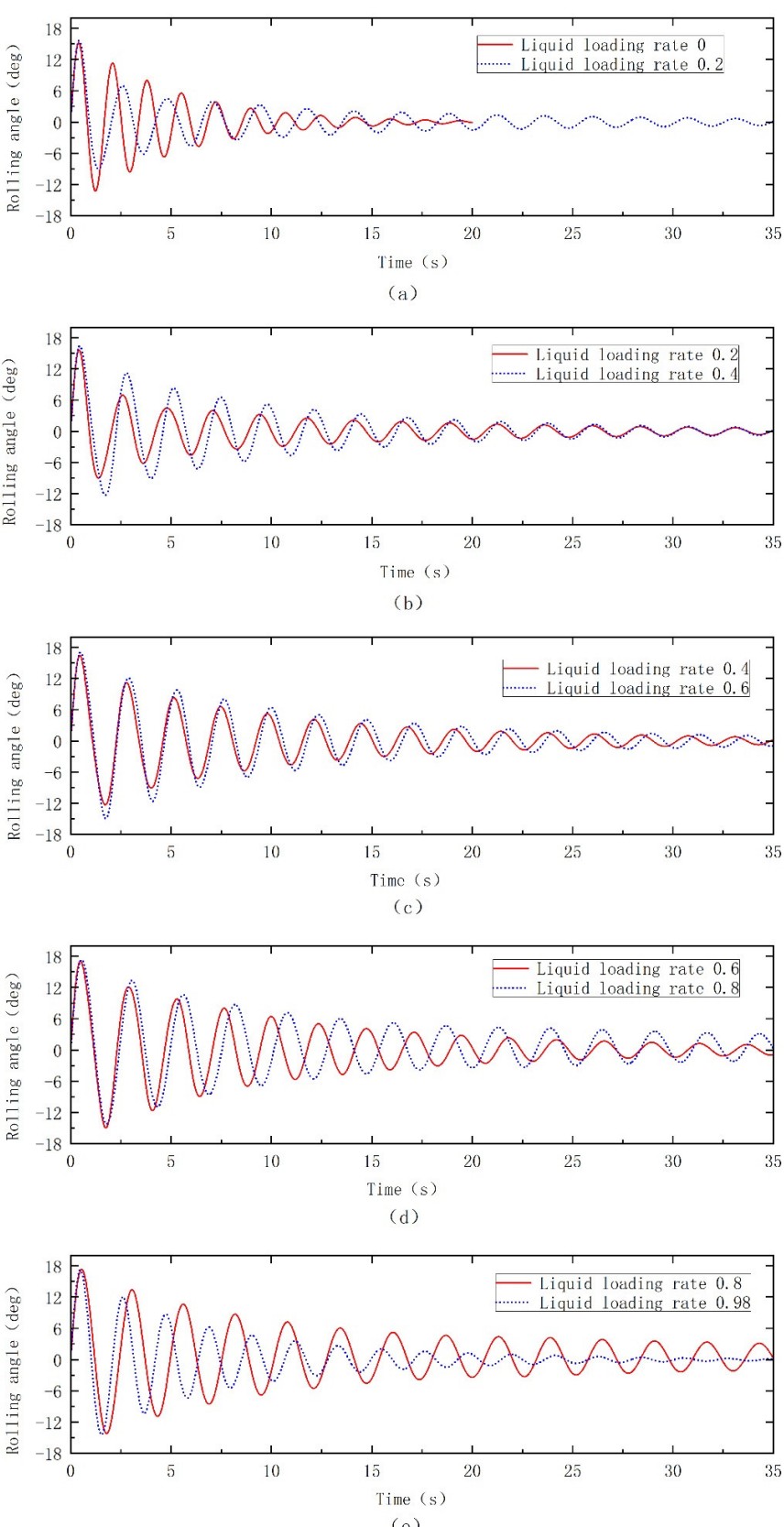

**Figure 15.** Rolling free decay curve of ships with different liquid loading rates. (**a**) Free decay curves for carrier ratios of 0 and 0.2, respectively; (**b**) Free decay curves for carrier ratios of 0.2 and 0.4, respectively; (**c**) Free decay curves for carrier ratios of 0.4 and 0.6, respectively; (**d**) Free decay curves for carrier ratios of 0.6 and 0.8, respectively; (**e**) Free decay curves for carrier ratios of 0.8 and 0.98, respectively.

Figure 16 shows the variation of the natural frequency of the ship's roll with the liquid loading rate. It can be seen that the sloshing of the tank reduces the natural frequency of the ship's roll, and the natural frequency of the ship's roll gradually decreases during the change of the liquid loading rate from 0.2 to 0.8. This can be explained from the moment curve in Figure 12: when the roll angle of the ship is not very small, the change of the sloshing moment of the tank is always opposite to the external wave moment, and the superposition of the two reduces the restoring moment of the ship. This results in reduced roll angular acceleration, thus longer roll period and lower natural frequency. The higher the liquid loading rate (0.2–0.8), the greater the sloshing moment of the tank, the smaller the restoring moment after superposition, and the smaller the roll angular acceleration, so the lower the natural frequency of roll. However, when the liquid loading rate is 0.98, the tank is basically in a fully loaded state. Figure 17 shows the change of liquid level in the tank under this liquid loading rate. Since the free liquid surface is close to the top of the tank, the sloshing moment of the tank will be at a lower level, which has little effect on the restoring moment. Therefore, the natural frequency of ship roll will increase.

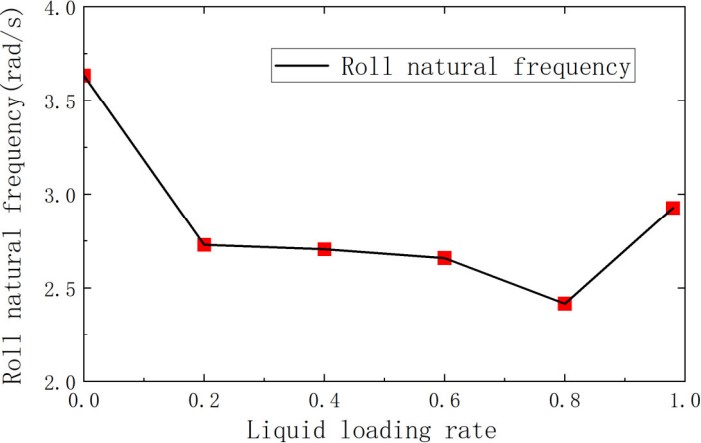

**Figure 16.** Variation curve of ship rolling natural frequency with liquid loading rate.

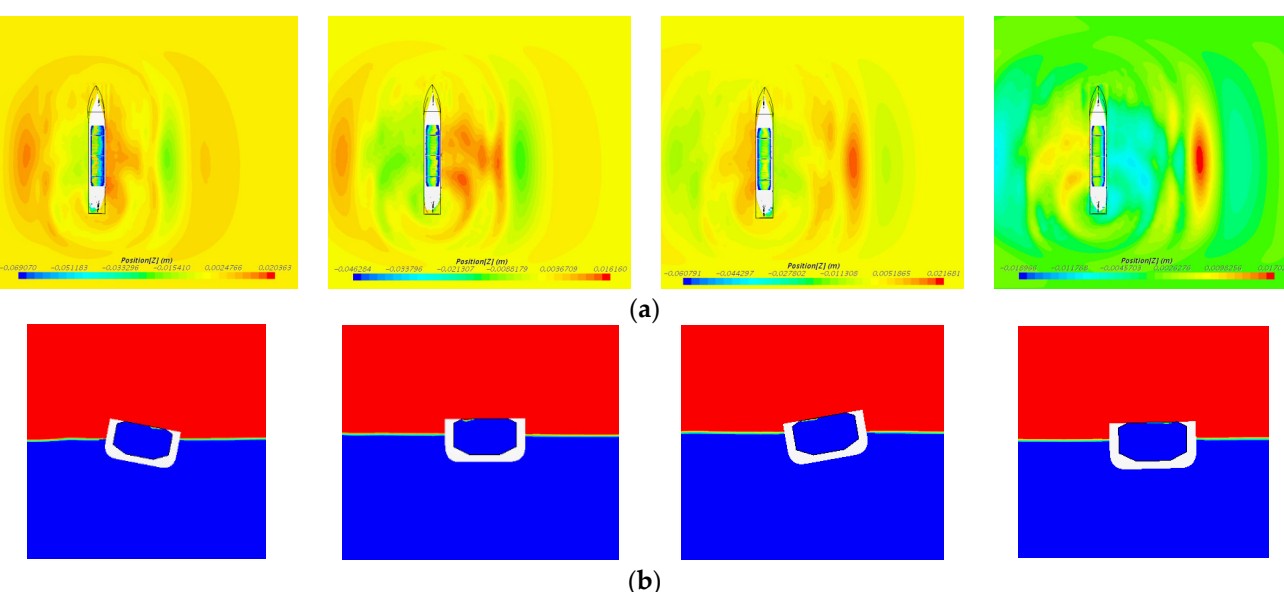

**Figure 17.** Rolling free decay motion of ship with a liquid loading rate of 0.98. (**a**) Vertical view; (**b**) midship section. In the picture, the liquid is blue and the gas is red.

*5.4. Influence of Wavelength of Incident Wave on Rolling Motion of Liquid-Carrying Ship*

Cases 1–4 in Table 4 calculate the rolling motion of the ship with a liquid loading rate of 0.8 under different incident wave wavelengths (3 m, 6 m, 11 m, 15 m). The influence of the incident wave wavelength on the rolling motion of the ship under this liquid loading rate is explored, and the calculation results are shown in Figure 18.

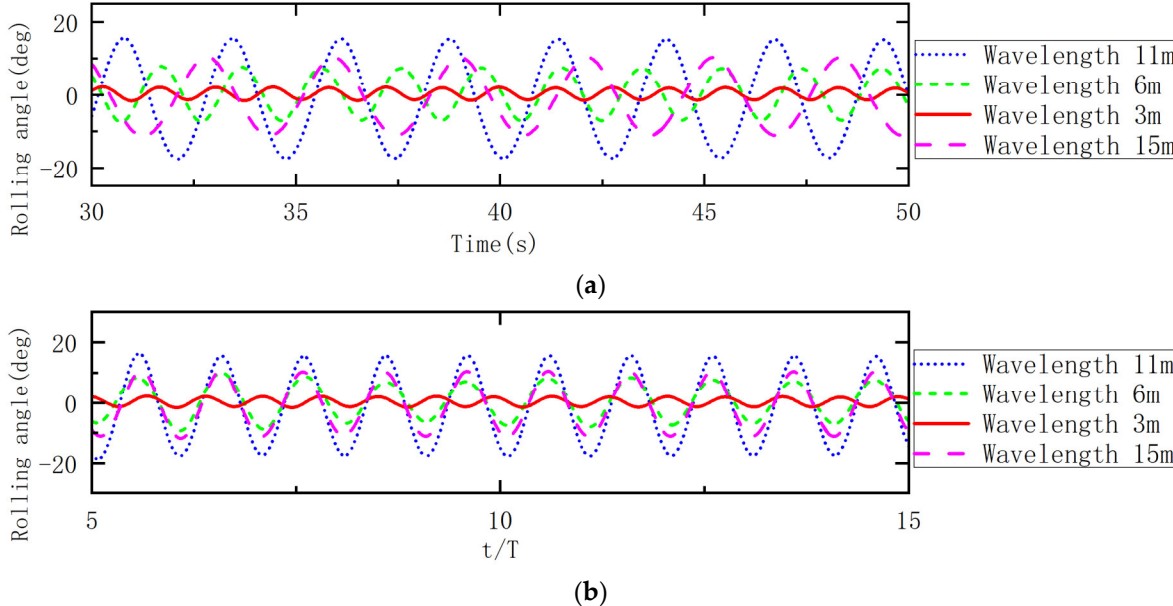

**Figure 18.** Time history curves of ship rolling under different wavelengths when the liquid loading rate is 0.8. (**a**) Time history curve of stable phase; (**b**) the result after time dimensionless.

It can be seen from Figure 18a that when the liquid loading rate is 0.8, the order of the roll angles in the stable stage corresponding to different incident wave wavelengths is: $\theta_{11} > \theta_{15} > \theta_6 > \theta_3$. This is mainly because the ratio of the incident wave frequency to the natural frequency of the ship's roll is different. It can be seen from Table 5 that the natural frequency of the ship's roll is 2.417 rad/s under this liquid loading rate. The ratio of the wave frequency corresponding to different incident wave wavelengths to the natural frequency of ship roll is shown in Table 6.

**Table 6.** The ratio of the incident wave frequency to the roll natural frequency when the loading rate is 0.8.

| Wavelength (m) | Incident Wave Frequency (rad/s) | Ratio of Incident Wave Frequency to Natural Frequency |
| --- | --- | --- |
| 3 | 4.532 | 1.875 |
| 6 | 3.205 | 1.326 |
| 11 | 2.367 | 0.979 |
| 15 | 2.027 | 0.838 |

It can be seen from Table 6 that the frequency of the wave corresponding to the incident wave wavelength of 11 m is closest to the natural frequency of the ship's roll, and the frequency of the wave corresponding to the incident wave wavelength of 3 m is the farthest from the natural frequency. Therefore, the roll amplitude of the ship is the largest when the incident wave wavelength is 11 m, and the roll amplitude is the smallest when the incident wave wavelength is 3 m.

Time dimensionless (the ratio of time to incident wave period) is shown in Figure 18b. It can be seen from the figure that under this liquid loading rate, when the wavelength of the incident wave is 6 m, 11 m, and 15 m, the rolling period of the ship is basically the same

as that of the incident wave. However, under the high frequency incident wave (when the wavelength is 3 m), the ship roll period is smaller than the incident wave period, that is, under the high frequency incident wave, the ship roll amplitude is smaller, but the speed is faster.

In Cases 5–8 and 9–12 in Table 4, the rolling motion law of the ship with the liquid loading rate of 0.4 and 0 under different incident wave wavelengths is studied, respectively. The calculation results of rolling under different loading rates are compared in Figure 19.

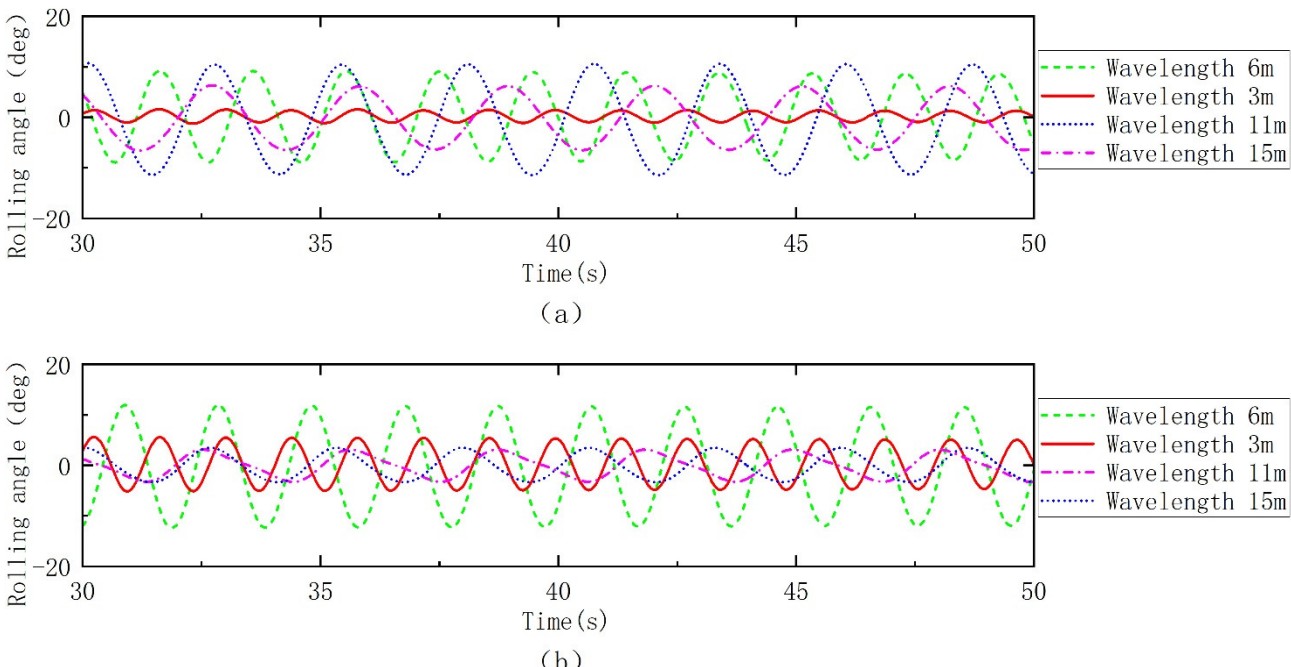

**Figure 19.** Time history curve of ship rolling motion under different incident wave wavelengths. (**a**) Liquid loading rate 0.4; (**b**) liquid loading rate 0.

It can be seen from Figure 19 that when the loading rate is 0.4, the order of the roll angles in the stable stages corresponding to different incident wave wavelengths is: $\theta_{11} > \theta_6 > \theta_{15} > \theta_3$; when the loading rate is 0, the stable stages corresponding to different incident wave wavelengths are in the following order: $\theta_6 > \theta_3 > \theta_{11} > \theta_{15}$. It can be seen that the order of roll angles corresponding to ships with different liquid loading rates is different, which is mainly due to the difference in the natural frequency of rolling of ships due to different liquid loading rates.

Tables 7 and 8 are the ratios of the incident wave frequency to the natural frequency of the ship's roll when the loading rate is 0.4 and 0, respectively. It can be seen that when the liquid loading rate is 0.4, the degree of closeness to the natural frequency of different incident wave wavelengths is from near to far: 11 m, 6 m, 15 m, 3 m. When the liquid loading rate is 0, the degree of closeness to the natural frequency of different incident wave wavelengths is from near to far: 6 m, 3 m, 11 m, 15 m.

**Table 7.** The ratio of the incident wave frequency to the roll natural frequency when the loading rate is 0.4.

| Wavelength (m) | Incident Wave Frequency (rad/s) | Ratio of Incident Wave Frequency to Natural Frequency |
|---|---|---|
| 3 | 4.532 | 1.675 |
| 6 | 3.205 | 1.185 |
| 11 | 2.367 | 0.875 |
| 15 | 2.027 | 0.749 |

**Table 8.** The ratio of the incident wave frequency to the roll natural frequency when the loading rate is 0.

| Wavelength (m) | Incident Wave Frequency (rad/s) | Ratio of Incident Wave Frequency to Natural Frequency |
|---|---|---|
| 3 | 4.532 | 1.247 |
| 6 | 3.205 | 0.882 |
| 11 | 2.367 | 0.651 |
| 15 | 2.027 | 0.558 |

*5.5. Influence of Liquid Loading Rate on Rolling Motion of Ship*

In 5.4, when the liquid loading rate is constant (the natural frequency of rolling is constant), the influence of different incident wave wavelengths on the rolling motion of the ship is analyzed. Next, when the wavelength of the incident wave is constant (the frequency of the incident wave is constant), the difference in the rolling motion of the ship with different liquid loading rates is analyzed.

Cases 1, 5, and 9 in Table 4 explore the law of ship rolling motion when the incident wave wavelength is 3 m and the liquid carrier rates are 0.8, 0.4, and 0, respectively. At this wavelength, the stability stage of the time history curve of the ship's rolling motion with different liquid loading rates is compared in Figure 20.

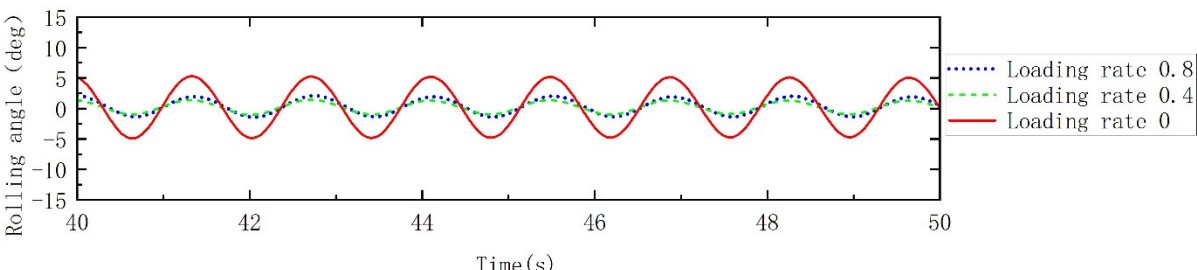

**Figure 20.** Rolling time history curves of ships with different liquid loading rates when the incident wave wavelength is 3 m.

From Figure 20, it can be seen that the amplitude of the roll angle changes with different liquid loading rates is different: the roll amplitude at 0 liquid loading rate is the largest, and the roll amplitude values when the liquid loading rate is 0.4 and 0.8 are close to and smaller than the roll amplitude value when the liquid loading rate is 0. It can be seen that under this wavelength of the incident wave, the roll angle of the ship becomes smaller after loading the tank with the liquid loading rate of 0.4 and 0.8. At this time, the tank sloshing in the two liquid-carrying states restrains the ship's rolling, and the tank sloshing is beneficial to the ship's rolling motion. This is mainly because the natural frequency of the rolling motion of the ship has changed after loading the tank. The natural frequency of ship roll with a liquid loading rate of 0 is the closest to the incident wave frequency, and the natural frequency of ship roll with a liquid loading rate of 0.4 and 0.8 is far from the incident wave frequency; see Table 9. The following conclusions can be drawn: the effect of tank sloshing on the ship's rolling motion is achieved by changing the natural frequency of rolling to make it different from the frequency of the incident wave.

In addition, cases 2, 6, and 10 in Table 4 explore the law of ship rolling motion when the incident wave wavelength is 6 m and the liquid carrier rates are 0.8, 0.4, and 0, respectively. Cases 3, 7, and 11 in Table 4 explore the law of ship rolling motion when the incident wave wavelength is 11 m and the liquid carrier rates are 0.8, 0.4, and 0, respectively. Cases 4, 8, and 12 in Table 4 explore the law of ship rolling motion when the incident wave wavelength is 15 m and the liquid carrier rates are 0.8, 0.4, and 0, respectively. Figure 21 compares the roll time history curves of ships with different liquid loading rates.

**Table 9.** Ratio of ship roll natural frequency to incident wave frequency when incident wave wavelength is 3 m.

| Liquid Loading Rate | Roll Natural Frequency (rad/s) | The Ratio of Natural Frequency to Incident Wave Frequency |
|---|---|---|
| 0.8 | 2.417 | 0.533 |
| 0.4 | 2.705 | 0.597 |
| 0 | 3.634 | 0.802 |

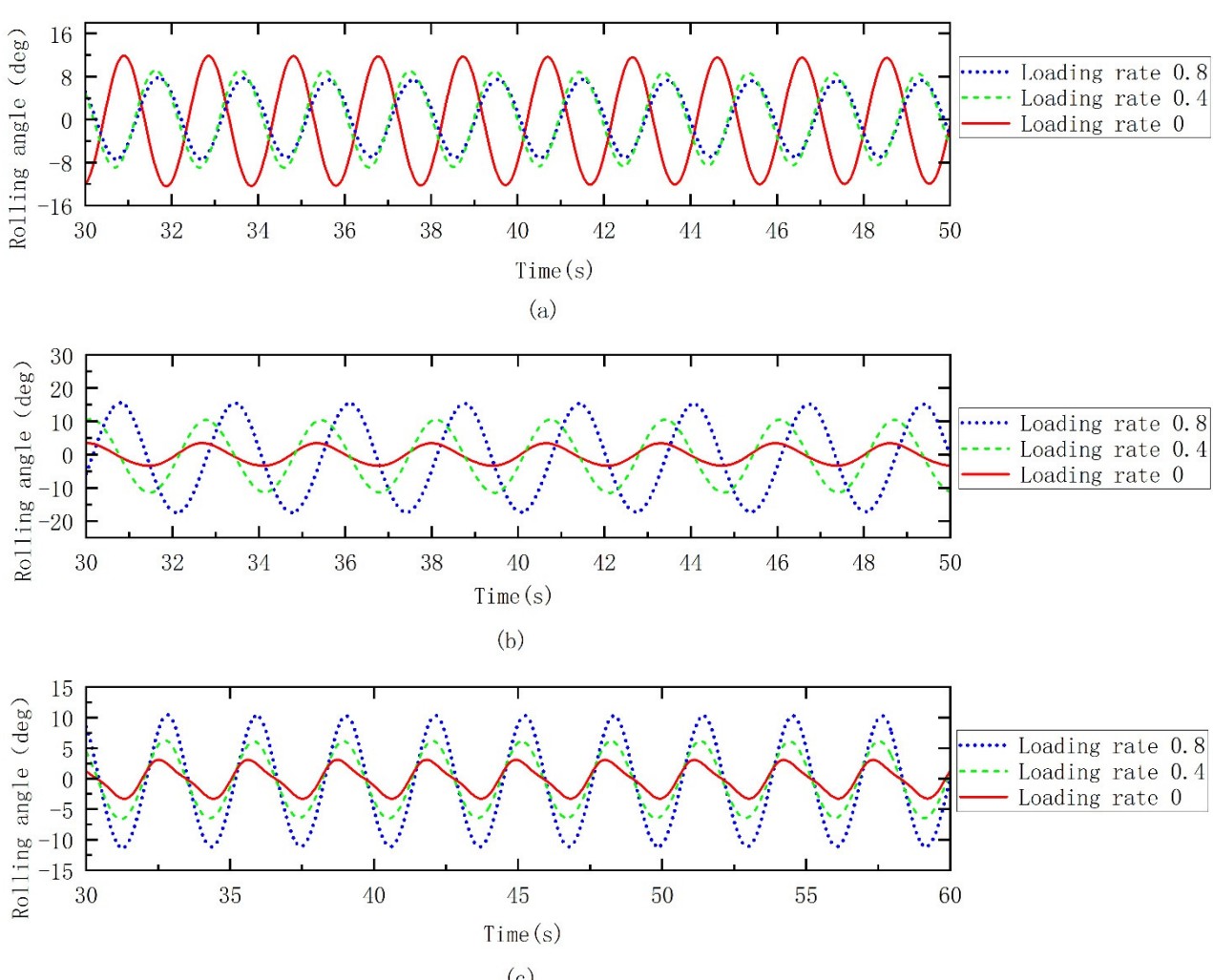

**Figure 21.** Rolling time history curves of ships with different liquid loading rates under different wavelengths. (**a**) Wavelength 6 m; (**b**) wavelength 11 m; (**c**) wavelength 15 m.

It can be seen from Figure 21a that when the wavelength of the incident wave is 6 m, the order of the roll angle of the ship with different liquid loading rates is: $\theta_0 > \theta_{0.4} > \theta_{0.8}$. Under this condition, the sloshing of the tank restrains the rolling motion of the ship. In addition, it can be seen from Figure 21b that when the wavelength of the incident wave is 11 m, the order of the ship roll angle with different liquid loading rates is: $\theta_{0.8} > \theta_{0.4} > \theta_0$. Under this condition, the sloshing of the tank promotes the rolling motion of the ship. It can be seen from Figure 21c that when the wavelength of the incident wave is 15 m, the order of the ship roll angle with different liquid loading rates is: $\theta_{0.8} > \theta_{0.4} > \theta_0$. Under this condition, the sloshing of the tank also promotes the rolling motion of the ship. Tables 10–12 analyze the ratio of the natural frequency of the ship's roll to the frequency of the incident wave.

**Table 10.** Ratio of ship roll natural frequency to incident wave frequency when incident wave wavelength is 6 m.

| Liquid Loading Rate | Roll Natural Frequency (rad/s) | The Ratio of Natural Frequency to Incident Wave Frequency |
| :---: | :---: | :---: |
| 0.8 | 2.417 | 0.754 |
| 0.4 | 2.705 | 0.844 |
| 0 | 3.634 | 1.134 |

**Table 11.** Ratio of ship roll natural frequency to incident wave frequency when incident wave wavelength is 11 m.

| Liquid Loading Rate | Roll Natural Frequency (rad/s) | The Ratio of Natural Frequency to Incident Wave Frequency |
| :---: | :---: | :---: |
| 0.8 | 2.417 | 1.021 |
| 0.4 | 2.705 | 1.143 |
| 0 | 3.634 | 1.535 |

**Table 12.** Ratio of ship roll natural frequency to incident wave frequency when incident wave wavelength is 15 m.

| Liquid Loading Rate | Roll Natural Frequency (rad/s) | The Ratio of Natural Frequency to Incident Wave Frequency |
| :---: | :---: | :---: |
| 0.8 | 2.417 | 1.192 |
| 0.4 | 2.705 | 1.334 |
| 0 | 3.634 | 1.793 |

The same conclusion can be drawn from the above three tables: the closer the incident wave frequency is to the natural frequency of the ship's rolling motion, the greater the amplitude of the ship's rolling motion will be. Therefore, it can be said that tank sloshing changes the amplitude of the ship's motion by changing the natural frequency of the ship's motion to make it different from the incident wave frequency.

## 6. Conclusions and Discussions

The purpose of the present research was to explore the effect of tank sloshing on the response to ship motion. Therefore, the pitching and rolling motions of ships with different liquid loading rates under different incident wave wavelengths were respectively calculated. The major results are summarized as follows:

1. In the head sea condition, the sloshing of the tank has little effect on the pitching motion of the ship. The main reason is that under this condition, the sloshing moment of the tank is much smaller than the external wave moment.
2. In the head sea condition, the wavelength of the incident wave has a great influence on the pitching motion of the ship. When the incident wave wavelength is close to the length of the ship, it is a dangerous situation because the pitching motion of the ship is the most violent (large amplitude and short period).
3. When the liquid loading rate is not very high, the sloshing of the tank will reduce the natural frequency of the ship's roll. The higher the liquid loading rate is, the lower the natural frequency of ship roll will be. This is mainly because the phase of the tank sloshing moment is always opposite to the external wave moment, which reduces the ship's restoring moment.
4. When the liquid loading rate is very high, since the free liquid surface is close to the top of the tank, the sloshing moment of the tank decreases, and its influence on the natural frequency of the ship's rolling motion decreases.
5. The sloshing of the tank has a significant effect on the rolling motion of the ship under the condition of transverse waves. The effect of tank sloshing on the rolling motion

of the ship is realized by changing the natural frequency of the rolling motion of the ship. The closer the incident wave frequency is to the natural frequency of ship roll, the greater the ship's roll amplitude will be.

In this paper, the CFD method is used to simulate the motion of a liquid-carrying ship under different waves. The results calculated by the numerical model are close to the experimental values, but some limitations exist. For example, the influence of tank bulkhead deformation and hull deformation is not considered. Further study is required to refine the model and make it closer to reality.

**Author Contributions:** X.F. made the computations, data analysis and wrote this paper; Z.H. did the proof reading. X.Z. guided the whole project. All authors have read and agreed to the published version of the manuscript.

**Funding:** This research was funded by the National Key Research and Development Program of China, grant number 2020YFB1506701, the National Natural Science Foundation of China, grant number 51739001; 51879051, Natural Science Foundation of Heilongjiang Province in China, grant number LH2020E071, and Open Fund of Zhejiang Provincial Key Laboratory of Wind Power Technology, grant number ZOE2020007.

**Institutional Review Board Statement:** Not applicable.

**Informed Consent Statement:** Not applicable.

**Data Availability Statement:** We choose to exclude this statement.

**Conflicts of Interest:** The authors declare no conflict of interest.

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
