# Peer review of "Research on Influence of Tank Sloshing on Ship Motion Response under Different Wavelengths"

_applsci, doi:10.3390/app12178647_

Round 1

Reviewer 1 Report

The manuscript titled " Research on influence of tank sloshing on ship motion response under different wavelengths " is interesting and well written. The organization of this manuscript is good and the analysis is accurate. The presented work can be accepted for publication if the following comments are addressed adequately:

Line-31: "...due to its fast and low-cost characteristics...". It depends on the size (mesh elements) of the computational domain, the treatment of the turbulence, and the models (VOF, multiphase...). In my opinion, that line should be rewritten in order to be more accurate.

Line-78: "...the k-e turbulence model". Which variant? (Std, RNG, realizable...). The wall treatment should be explained. The y+ parameter is very important and should be given.

Line-100: "... the mesh quality of the overlapping mesh is BETTER...". In my opinion, the overlapping grid technology may be more convenient and it is easier to mesh. Althought, why is the mesh quality better? Is it a consequence of the fact that this technology does not require mesh deformation?

Line-118: "...to eliminate waves at the boundary of the pool...". Is the wave reflection completely eliminated? The effect of the size of the computational domain should be studied/shown.

Eq. 10: The forcing coefficent should be given.

Line-125: "..., try to ensure..." This line should be rewritten.

Line-133..134: The size of the time step should be given. In addition, the number of iterations per time step is also missing.

The Courant Number is missing.

What convergence criteria were chosen?

Table 1.
    * 230 / 37.9 = 6.0686 ~ 6.07 != 6.0702
    * 32.2 / 37.9 = 0.8496 != 0.8598.
Is it correct?

Line-169: (4) "The grid size of the overlappling..." It is very interesting. Have you performed an analysis of the maximum difference between the size of the elements in both areas? Is a 2:1 ratio too large?

Grid independence analysis is missing.

The numerical model is missing: Solver (pressure based / density based), gradients, spatial discretization, temporal discretization, velocity-pressure coupling, VOF/multiphase/...

Figure 10:
        * For comparison purposes, the three contours should have the same range on the colormap.
        * The quality should be improved (the text is too small and blurry)

Figure 16+1:
        * The caption is missing.
        * The colormap is missing.

Line 484:
        * The CFD method is based on viscous flow. What does this mean?

DOI, crossref, or URL should be included at the end of each entry in the reference list.

Author Response

Many thanks for the comments of reviewer 1, more details of the responses to the reviewer 1 can be checked in the attachment.

Reviewer 2 Report

Comment 1:The articles that are cited in the manuscript are too old. The most recent of the cited references is 2019. Authors need to update the introduction section with a recent publication.
Authors may consider the following related articles.

10.9734/JAMCS/2017/36489 https://journaljamcs.com/index.php/JAMCS/article/view/24042

Comment 2 The manuscript needs to be proofread to remove some typographic errors.

Author Response

Many thanks for the comments of reviewer 2, more details of the responses to the reviewer 2 can be checked in the attachment.

Reviewer 3 Report

In this paper the motion of the ship model with tanks at different wavelengths are simulated based on the CFD software. Also, the pitching motion and rolling motion of the liquid carrier in the head sea condition and the transverse wave condition are simulated respectively. This manuscript is valuable for publication in this journal and its writing in good language. However, before publication, I suggest the authors answer these questions.

1-      In abstract section, the authors need to add the main outcomes as a numbers.

2-      In abstract section, however, the word of CFD software is general task weather the authors work on a specific term of CFD. Thus, kindly the authors need to add the exactly CFD software methods which they used with its brief conditions.

3-      In introduction section line#34, the authors mention the experimental research method, however, they did not present those authors who works on experimental part and their found and where is the lack!

4-      Kindly, this manuscript includes many symbols and need to present it in separate table.

5-      What are the parameters or the conditions that the authors want to improved it in this study? Its not clear for me.

6-      The author needs to take the impact of the viscosity index inside the tank? Weather its Newtonian or non-Newtonian fluid. And can this parameter play an important role to evaluate the tanker design and carry weight as well as the movements of tanker?

7-      The author did not mention if this kind of CFD analysis take the impact of friction between the tanker body and the liquid waves (water), which may effect on the age of the metal body (tanker body).

8-      In section Conclusions and Discussions line#487, “Further study is required to refine the model and make it closer to reality.” This kind of research required extensive experimental work with online real data in order to compere it to any theoretical models and simulation software as well.

Author Response

Many thanks for the comments of reviewer 3, more details of the responses to the reviewer 3 can be checked in the attachment.

Round 2

Reviewer 1 Report

All comments have been addressed in the revision.